

# A hybrid method for heartbeat classification via convolutional neural networks, multilayer perceptrons and focal loss

Tao Wang[1], Changhua Lu[1], Mei Yang[2], Feng Hong[1] and Chun Liu[3]

[1] School of Computer and Information, Hefei University of Technology, Hefei, Anhui, China
[2] Beijing Huaru Technology Co., Ltd. Hefei Branch, Hefei, Anhui, China
[3] School of Electrical Engineering and Automation, Hefei University of Technology, Hefei, Anhui, China

## ABSTRACT

**Background:** Heart arrhythmia, as one of the most important cardiovascular diseases (CVDs), has gained wide attention in the past two decades. The article proposes a hybrid method for heartbeat classification via convolutional neural networks, multilayer perceptrons and focal loss.

**Methods:** In the method, a convolution neural network is used to extract the morphological features. The reason behind this is that the morphological characteristics of patients have inter-patient variations, which makes it difficult to accurately describe using traditional hand-craft ways. Then the extracted morphological features are combined with the RR intervals features and input into the multilayer perceptron for heartbeat classification. The RR intervals features contain the dynamic information of the heartbeat. Furthermore, considering that the heartbeat classes are imbalanced and would lead to the poor performance of minority classes, a focal loss is introduced to resolve the problem in the article.

**Results:** Tested using the MIT-BIH arrhythmia database, our method achieves an overall positive predictive value of 64.68%, sensitivity of 68.55%, $f1$-score of 66.09%, and accuracy of 96.27%. Compared with existing works, our method significantly improves the performance of heartbeat classification.

**Conclusions:** Our method is simple yet effective, which is potentially used for personal automatic heartbeat classification in remote medical monitoring. The source code is provided on https://github.com/JackAndCole/Deep-Neural-Network-For-Heartbeat-Classification.

Corresponding authors
Tao Wang, wtustc@mail.ustc.edu.cn
Chun Liu, dqlch03@hfut.edu.cn

## INTRODUCTION

Heart arrhythmia, one of the most important cardiovascular disease (CVD), refers to the irregular beating of the patient's heart. Most arrhythmias are asymptomatic and not severe, but some could cause heart disease symptoms such as passing out, lightheadedness, chest pain, shortness of breath, and even stroke and cardiac arrest such as ventricular fibrillation, ventricular escape and atrial fibrillation, which are extremely dangerous and

need immediate treatment. According to statistics from the World Health Organization, the number of CVD deaths in 2015 is close to 17.7 million, accounting for about 31% of the total deaths (*Shen et al., 2019*).

Electrocardiogram (ECG), a device that records the electrical activity of the heart, is widely used to diagnose cardiac arrhythmias in clinical (*Mondéjar-Guerra et al., 2019*). An ECG signal consists of a series of periodically repeating heartbeats. Each heartbeat usually contains a QRS complex, a T wave, and a P wave, in a few cases there is a U wave (*Vulaj et al., 2017*). The most significant characteristic of an ECG signal is the QRS complex. By analyzing this complex, arrhythmia can be detected. However, the occurrence of arrhythmia is intermittent, especially in the early stages, which makes it difficult to perform effective detection in a short time (*Mondéjar-Guerra et al., 2019*).

To solve this problem, a Holter monitor is often used to collect long-term heart electrical activity recordings (*Sannino & De Pietro, 2018*). In general, an ECG recording lasts several minutes or even hours. Investigating a variety of abnormal arrhythmias beat-by-beat from long-term ECG recordings is very exhausting, even for trained cardiologists. Therefore, there is an urgent need for a computer-aided method to automatically detect abnormal heartbeats from long-term ECG data.

Over the past two decades, a lot of research works (*De Albuquerque et al., 2018*; *De Chazal, O'Dwyer & Reilly, 2004*; *Mondéjar-Guerra et al., 2019*) have been spent on classifying heartbeats automatically. Most of these methods are based on morphological characteristics of heartbeats and traditional signal processing techniques. However, the ECG waveform and its morphological characteristics (e.g., the shape of the QRS waves and P wave) of different patients are significantly different, and for the same patient, there are also differences in different circumstances (*Mondéjar-Guerra et al., 2019*), so the fixed features used in these methods are not sufficient to accurately distinguish arrhythmias for all patients. In recent years, some deep neural networks have been proposed, such as convolutional neural networks (CNN), which can automatically extract morphological features and adapt to variations between patients.

Nevertheless, there is another challenge when processing medical data. Due to the limited number of rare classes, the number of one class may greatly exceed that of other classes, that is, the distribution of classes is imbalanced. However, most algorithms try to minimize the overall classification loss during the training process, which implies that these classes are equally important and the same misclassification cost is allocated to all types of errors. As a result, the classifier will tend to correctly classify and favor more frequent classes.

The article presents a hybrid method for heartbeat classification via CNN, multilayer perceptrons (MLP) and focal loss. An overall structure of the method is displayed in Fig. 1. The morphological features are extracted by one-dimensional (1D) CNN and combined with the RR intervals features as the input of MLP. The RR intervals features contain the dynamic information of the heartbeat, which could help better capture the pattern of the ECG waveform. Furthermore, considering that the heartbeat classes are

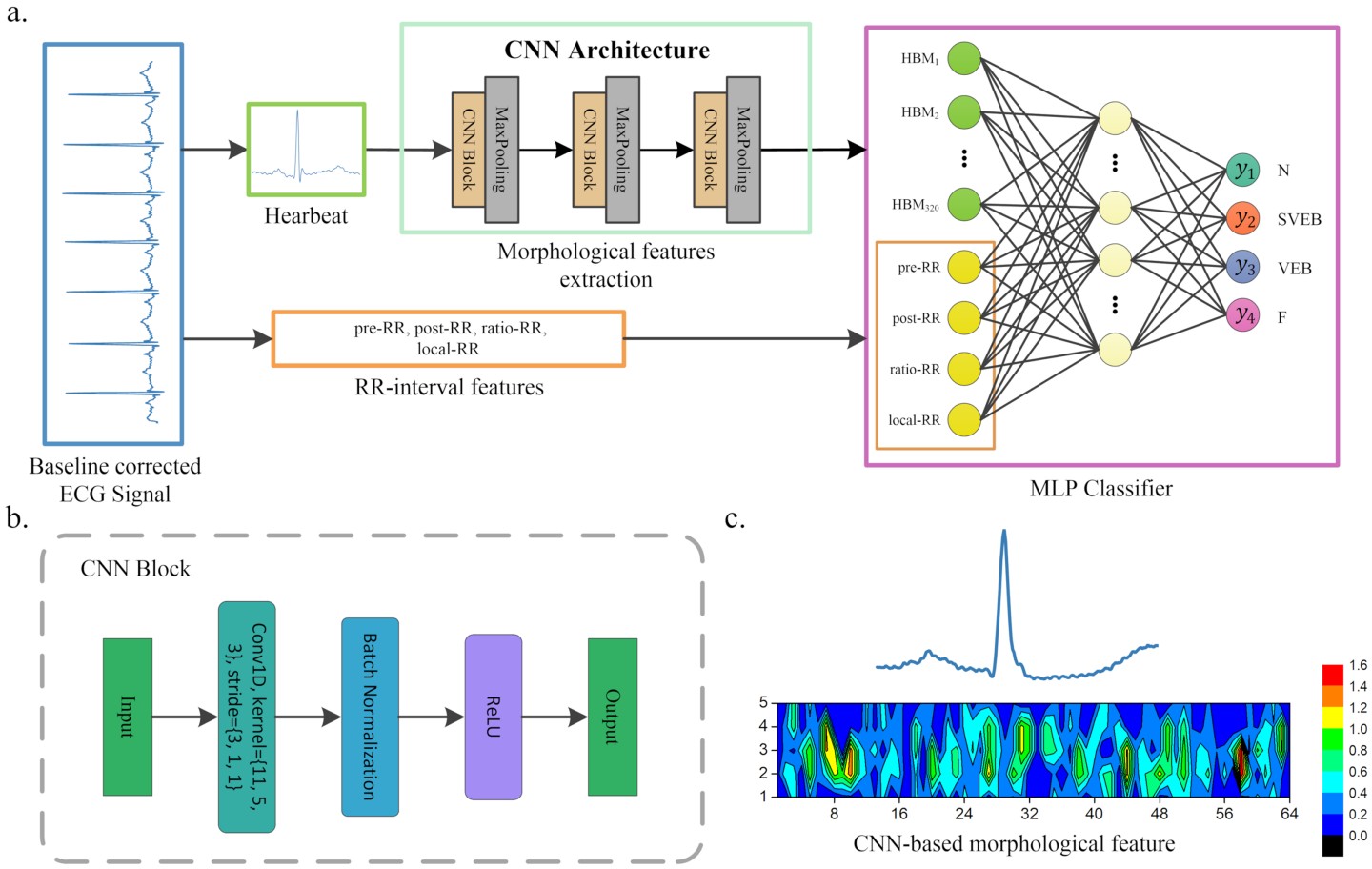

**Figure 1 A scheme of our proposed method.** (A) Overview of our method. (B) CNN block of our CNN architecture. (C) An example of morphological features extracted by CNN, where the upper part heartbeat signal is the input of CNN, and the lower part is the features extracted by CNN, that is, the morphological features. These features will be flattened and combined with the RR interval features when used as the input of the MLP classifier.

imbalanced and would lead to the poor performance of minority classes, a focal loss is introduced to solve the problem. It shows superior performance in various application environments (*Howland et al., 2002*; *Lin et al., 2017*; *Zhou, Waterman-Storer & Cohan, 2002*). By testing in the well-known MIT-BIH arrhythmia database (*Moody & Mark, 2001*), our method achieves superior classification performance than existing heartbeat classification methods. Note that the accuracy of the ECG classification method has been standardized according to the Association for the Advancement of Medical Instrumentation's (AAMI) recommendations. The proposed method obtains an overall PPV of 64.68%, SE of 68.55%, F1 of 66.09%, and accuracy of 96.27%.

The article is organized as follows: "Related Works" presents the related works of heartbeat classification. The proposed method and loss function are introduced in "Methods" and "Loss Function". The dataset and the performance of our method against existing works are described in "Results". "Discussion" discusses the conclusions.

## RELATED WORKS

The existing automatic heartbeat classification works can be divided into two paradigms: intra-patient paradigm and inter-patient paradigm (*De Chazal, O'Dwyer & Reilly, 2004*; *Sannino & De Pietro, 2018*). In the intra-patient paradigm, the dataset is based on the heartbeat label split into training and test subsets, so an ECG recording will appear in two subsets (*Sannino & De Pietro, 2018*). According to *De Chazal, O'Dwyer & Reilly (2004)*, the results of this paradigm are biased, resulting in an accuracy of about 100% in the test phase, because the patient's characteristics are learned during the training phase (*Sellami & Hwang, 2019*). However, in actual scenarios, the trained model must be able to handle inter-patient variations during the training phase.

In the inter-patient paradigm, the training set and test set are from different patients (*Sannino & De Pietro, 2018*), so the differences between patients will be considered during the training process. The classifier will show a better generalization capability. For instance, *De Chazal, O'Dwyer & Reilly (2004)* propose a linear discriminant heartbeat classification method based on heartbeat morphological and dynamic features. Their method achieves a PPV of 38.5%, SE of 75.9% in the SVEB class, and a PPV of 81.6%, SE of 80.3% in the VEB class. *Ye, Kumar & Coimbra (2012)* apply wavelet transform and independent component analysis (ICA) to extract morphological features from heartbeats, and combined with dynamic RR interval features develop an support vector machine (SVM) method to classify heartbeat. A PPV of 52.3%, SE of 60.8% in the SVEB class, and a PPV of 63.1%, SE of 81.5% in the VEB class are obtained by their method. However, the classification accuracies of these methods are significantly lower than the intra-patient paradigm-based methods. This is due to variations of ECG characteristics between patients.

Recently, with the rapid development in deep learning, deep neural networks-based, especially CNN-based, heartbeat classification methods have received a lot of attention. For example, *Yıldırım et al. (2018)* develop a 1D-CNN for arrhythmia detection based on long-term ECG signal. Their method achieves 91.33% overall accuracy in 17 cardiac arrhythmias. Similarly, *Sellami & Hwang (2019)* develop a CNN with a batch-weighted loss function for heartbeat classification. *Hannun et al. (2019)* present a deep neural network with residual block to classify 12 rhythm classes. *Romdhane et al. (2020)* based on CNN and focal loss propose an ECG heartbeat classification method. Although the performance of heartbeat classification is improved, these works mainly focus on using CNN to extract the heartbeat morphological features, while ignoring the influence of RR intervals on heartbeat classification. Research shows that by integrating RR interval features, the performance of heartbeat classification can be significantly improved (*De Chazal, O'Dwyer & Reilly, 2004*; *Mondéjar-Guerra et al., 2019*; *Sannino & De Pietro, 2018*). *Romdhane et al. (2020)* try to use an improved heartbeat segmentation method to make CNN capture RR interval information, but in their work, CNN can only extract the previous RR interval information at most. This is due to the incomplete division of the right interval. Different from existing works, we pre-extract the RR interval

information in advance, and then combine it with CNN-based morphological features as the input of the classifier.

In addition to the above two classification paradigms, a hybrid paradigm has also been studied by some scholars, namely patient-specific paradigm. In the patient-specific paradigm, a global model is first built and then use part of patient data to tune the model to form a local model. *De Chazal, O'Dwyer & Reilly (2004)* shows that this paradigm is superior to a pure inter-patient model. However, this paradigm requires a professional doctor to label part of the ECG data, and an engineer to fine-tuning the model in clinical. Meanwhile, the patient's ECG signal may change significantly over time, that is, the current ECG signal may undergo large variations at some time in the future, and the use of a previously fine-tuned local classifier may lead to larger misclassification. We focus on the performance of our method in the inter-patient paradigm in the article.

## METHODS

Figure 1 shows the overall structure of the proposed method. The proposed method includes three steps: ECG denoising, feature extraction, and classification. The feature extraction step contains RR intervals features extraction and morphological features extraction via CNN architecture.

### ECG denoising

The ECG signal is usually disturbed by various noises such as electromyography noise, power line interference and baseline wandering (*Chen et al., 2017*), which makes useful features to be difficultly extracted. In this step, most previous works typically perform a baseline wandering removal and then high-frequency noise filtering (*Mondéjar-Guerra et al., 2019*). However, excessive filtering will lead to the loss of some helpful information in the ECG signal. Since CNN has better noise immunity (*Huang et al., 2018*), we only perform baseline wandering removal and preserve as much information as possible from the raw ECG signal.

Two median filters are combined to remove the baseline wandering of the ECG signal in the article. First, the QRS complexes and P-waves are removed using a 200-ms width median filter, and then a 600-ms width median filter is further adopted to remove T-waves. The output is the baseline wandering of the ECG signal, and the baseline-corrected ECG signal can be achieved by subtracting it from the original signal. An effect of baseline wandering removal is shown in Fig. 2.

After obtaining the baseline-corrected ECG signal, the ECG is further segmented into a series of heartbeats based on the labeled R-peaks. In specific, for each heartbeat, we obtain 200 sampling points of the ECG signal segment, 90 sampling points before and 110 sampling points after the labeled R peak. The R-peak detection is not the focus of the article and we directly use labeled R-peaks in the dataset, as there are many high-precision (>99%) R-peak detection methods in the literature (*Gacek & Pedrycz, 2012*; *Pan & Tompkins, 1985*).

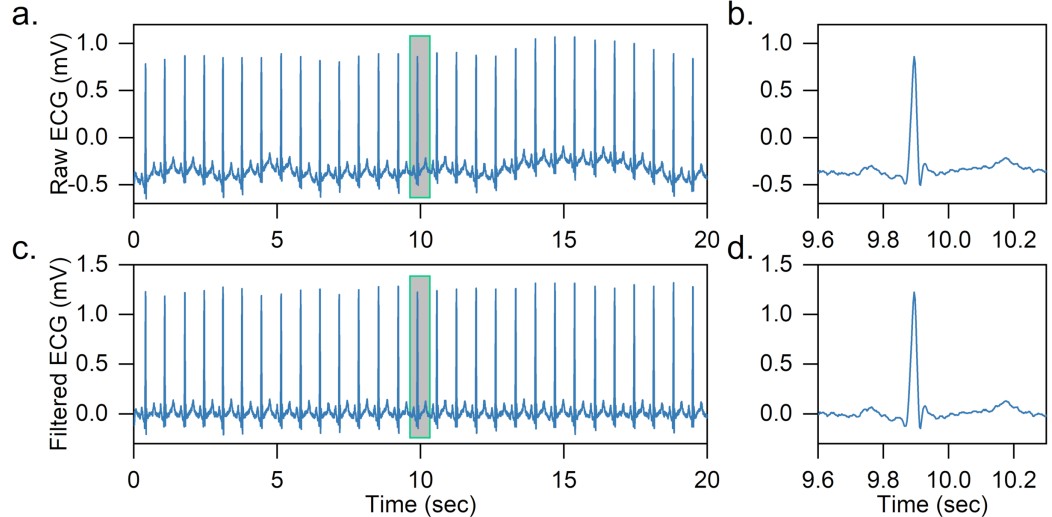

**Figure 2 Example of baseline wandering removal.** (A) Raw ECG signal. (B) Raw heartbeat. (C) Baseline-corrected ECG signal. (D) Baseline-corrected heartbeat. It is easy to notice that after removing the baseline wandering, the heartbeat is shifted to zero.

## RR interval features extraction

The time interval between two consecutive R-peaks is normally called the RR interval (*Ruangsuwana, Velikic & Bocko, 2010*), which contains the dynamic information of the heartbeat. To capture this information for heartbeat classification, four features are extracted from the RR interval, namely previous RR interval, post RR interval, ratio RR and local RR interval. The previous RR interval refers to the distance between the current R-peak position and the previous R-peak, the post RR interval is the distance between the current R-peak position and the following one. The ratio RR represents the ratio of the previous RR interval and the post RR interval. These three features reflect the instantaneous rhythm of a heartbeat. The average value of the 10 RR intervals before the current heartbeat is taken as the local RR interval, which represents the overall rhythm in the past. Due to the inter-patient variations in the ECG signal, the RR interval of different patients cannot be directly compared, in this article we use the entire patient's ECG signal to calculate the average RR interval, and subtract it from all RR characteristics (expect the ratio RR) to eliminate this effect.

## Morphological features extraction via CNN architecture

Convolutional neural networks is a powerful deep neural network inspired by visual neuroscience (*Chu, Shen & Huang, 2019b*). It has been successfully used in speech recognition, natural language processing, image classification, and biomedical signal (*Palaz & Collobert, 2015*; *Pourbabaee, Roshtkhari & Khorasani, 2018*; *Yin et al., 2017*). Given an image, CNN can effectively learn high-level abstractions, which can then be input into the classifier (e.g., fully connected neural network and SVM) for classification (*Zhang, Zhou & Zeng, 2017*). A CNN usually consists of convolutional layers, activation functions, and pooling layers, and sometimes including batch normalization layers.

Convolutional Layer: It is the most important component in CNN and performs convolution operation on the input data (*Liu & Chen, 2017*). Let $f_k$ and $s$ be the filter and the 1D ECG signal, respectively. The output of the convolution is calculated as follows:

$$C[i] = s(i) * f_k(i) = \sum_m s(m) f_k(i - m)$$

where $m$ is the size of the filter and the filter $f_k$ is realized by sharing the weights of adjacent neurons.

Activation Function: The activation function is used to determine whether the neuron should be activated. The purpose is to enable neurons to achieve nonlinear classification. Rectifier Linear Unit (ReLU) is one of the most widely used activation function, which can be expressed as

$$f(x) = \max(0, x)$$

where $x$ is the output value of the neuron.

Pooling layer: The pooling layer, also known as the down-sampling layer, is an operation that decreases the computational intensity by reducing the output neuron dimension of the convolutional layer, and can handle some variations due to signal shift and distortion (*Zhang, Zhou & Zeng, 2017*). The most widely used pooling method is the max-pooling, which is to apply the maximum function over input $s$. Let $m$ be the filter size, and the output is:

$$M(x) = \max\left\{ s(x + k) \| k | \leq \frac{m - 1}{2} \right\}$$

Batch Normalization Layer: The batch normalization layer is a technology for standardizing network input, applied to either the activations of a prior layer or inputs directly, which can accelerate the training process, and provides some regularization, reducing generalization error. Let $B = \{x_i, i = 1, \cdots, m\}$ be a mini-batch of the entire training set, the output of batch normalization is as follows:

$$\widehat{x_i} = \frac{x_i - \mu_B}{\sqrt{\sigma_B^2 + \varepsilon}}$$

where $\sigma_B$ and $\mu_B$ are the variance and the mean of training set $B$, respectively. $\varepsilon$ is an arbitrarily small constant to ensure the denominator is not zero.

A CNN is developed and utilized for heartbeat morphological feature extraction in this article. The CNN architecture is displayed in Fig. 1. It contains three convolutional blocks and three pooling layers. Each convolutional block includes a convolution layer, a ReLU activation function and a batch normalization layer. The kernel of the convolution is reduced as the network structure becomes deeper. For instance, the first convolution kernel is 11, while the second is reduced to 5. A batch normalization and ReLU activation are applied after each convolution operation, and a max-pooling is used to reduce the spatial dimension. Note that the parameters of the convolutional network are usually set based on the author's experience. The detailed parameters of CNN architecture are listed in Table 1. The output of the last pooling layer is the morphological features

**Table 1 The detailed parameters of our proposed deep neural network.**

| Layers | Layer name | Kernel size | No. of filters | Stride | Output shape | No. of trainable parameters | No. of non-trainable parameters |
|---|---|---|---|---|---|---|---|
| 0 | Input1[a] | – | – | – | $200 \times 1$ | – | – |
| 1 | 1D Convolution | 11 | 16 | 3 | $64 \times 16$ | 192 | – |
| 2 | Batch Normalization | – | – | – | $64 \times 16$ | 32 | 32 |
| 3 | ReLU | – | – | – | $64 \times 16$ | – | – |
| 4 | Max-Pooling | 3 | – | 2 | $31 \times 16$ | – | – |
| 5 | 1D Convolution | 5 | 32 | 1 | $27 \times 32$ | 2,592 | – |
| 6 | Batch Normalization | – | – | – | $27 \times 32$ | 64 | 64 |
| 7 | ReLU | – | – | – | $27 \times 32$ | – | – |
| 8 | Max-Pooling | 3 | – | 2 | $13 \times 32$ | – | – |
| 9 | 1D Convolution | 3 | 64 | 1 | $11 \times 64$ | 6,208 | – |
| 10 | Batch Normalization | – | – | – | $11 \times 64$ | 128 | 128 |
| 11 | ReLU | – | – | – | $11 \times 64$ | – | – |
| 12 | Max-Pooling | 3 | – | 2 | $5 \times 64$ | – | – |
| 13 | Flatten | – | – | – | 320 | – | – |
| 14 | Input2[b] | – | – | – | 4 | – | – |
| 15 | Concatenate | – | – | – | 324 | – | – |
| 16 | Dense | – | – | – | 64 | 20,800 | – |
| 17 | Dense | – | – | – | 4 | 260 | – |

**Notes:**
[a] Refers to the raw signal of the heartbeat. The morphological features of the heartbeat will be obtained through the CNN architecture.
[b] Is the RR interval features of the heartbeat. It will be combined with the CNN-based morphological features to build the final classification model.

extracted by CNN from the heartbeat. An illustration of the morphological features extracted from the heartbeat is shown in Fig. 1C.

## MLP classifier

The CNN-based morphological features and RR interval features are combined as the input of the classifier in the article. In general, any classifier (i.e., SVM and random forest (RF)) can be used for heartbeat classification. Here, we adopt a multilayer perceptron (MLP, also known as fully connected neural networks in deep learning) as the classifier. The reason behind this is that CNN and MLP can be combined for parameter training (we call it one-step training). Compared with other methods, this usually achieves better performance. Specifically, our MLP classifier contains an input layer, a hidden layer and an output layer. The input layer consists of two parts of information: CNN-based morphological features and RR interval features. The hidden layer has 64 neurons, and each neuron is connected to the input features. The output layer neurons are 4 in the article, each representing a kind of arrhythmia or normal heartbeat. The details of our method are shown in Fig. 1 and Table 1.

## Loss function

Before training a deep neural network, a loss function is first needed. The cross-entropy loss is the most widely used in deep neural network classification (*Chu, Wang & Lu, 2019a*).

However, this loss function does not address the class imbalance problem. A focal loss function is introduced in the article to deal with this problem.

## Cross-entropy loss

The cross-entropy is a measure in information theory (*Robinson, Cattaneo & El-Said, 2001*). It is based on entropy and calculates the difference between two probability distributions. Closely related to KL divergence that computers the relative entropy between two probability distributions, but the cross-entropy calculates the total entropy between the distributions. The cross-entropy is usually taken as the loss function in deep neural network classification (*Chu, Wang & Lu, 2019a*).

Let $t_i$ and $p_i$ be the ground truth and the estimated probability of each category, the cross-entropy loss is computed by:

$$CE = -\sum_{i}^{C} t_i \cdot \log(p_i)$$

where $C$ refers to the category set of the heartbeat. In the cross-entropy loss, each category is treated equally, which causes the majority category to overwhelm the loss and the model tends to classify to the majority category in an imbalanced environment.

## Focal loss

A characteristic of cross-entropy loss is that even easy-to-classify examples can cause significant losses, which will cause the loss of easy examples that constitute most of the dataset during the training process to negatively affect rare classes (*Lin et al., 2017*). The focal loss is designed to deal with this imbalanced problem by reshaping the cross-entropy loss function by reducing the attention to easy examples and focusing on difficult ones. A general formula for focal loss is expressed as:

$$FL = -\sum_{i}^{C} t_i \cdot (1 - p_i)^{\gamma} \log(p_i)$$

where $\gamma$ acts as the modulating factor. As shown in Fig. 3, the higher the $\gamma$ value, the lesser the cost incurred by well-classified examples. In practice, the α-balanced variant of the focal loss is usually used when one or more categories are highly imbalanced, which is defined as:

$$FL' = -\sum_{i}^{C} t_i \cdot \alpha_i (1 - p_i)^{\gamma} \log(p_i)$$

where $\alpha_i$ is the weighting factor of each category.

## RESULTS

### Data set

The ECG dataset from the MIT-BIH arrhythmia database (*Moody & Mark, 2001*) is used to test our proposed method. This dataset contains 48 30-min ambulatory two leads ECG signal records collected from 47 subjects. Each ECG signal is sampled at 360 Hz with

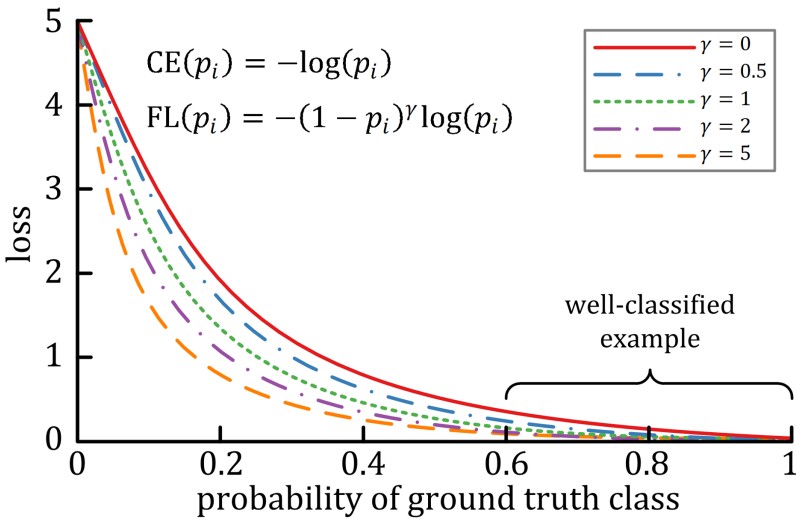

**Figure 3** The relationship between the modulating factor γ and the cost of the well-classified examples (*Lin et al., 2017*).               

**Table 2** Mapping of AAMI classes and MIT-BIH arrhythmia heartbeat types.

| AAMI classes | MIT-BIH types | MIT-BIH annotate |
|---|---|---|
| Normal (N) | Normal beat (NOR) | N |
| | Nodal (junctional) escape beat (NE) | j |
| | Atrial escape beat (AE) | e |
| | Right bundle branch block beat (RBBB) | R |
| | Left bundle branch block beat (LBBB) | L |
| Supraventricular ectopic beat (SVEB) | Aberrated arial premature beat (aAP) | a |
| | Premature or ectopic supraventricular beat (SP) | S |
| | Nodal (junctional) premature beat (NP) | J |
| | Atrial premature beat (AP) | A |
| Ventricular ectopic beat (VEB) | Ventricular escape beat (VE) | E |
| | Premature ventricular contraction (PVC) | V |
| Fusion beat (F) | Fusion of ventricular and normal beat (fVN) | F |
| Unknown beat (Q) | Unclassifiable beat (U) | Q |
| | Fusion of paced and normal beat (fPN) | f |
| | Paced beat (P) | / |

an 11-bit resolution. The first lead is the modified-lead II (ML II), and the second lead depends on the record, one of V1, V2, V4 or V5. The heartbeat of these ECG signals is independently labeled by two or more doctors, and there are about 110,000 heartbeats.

According to the recommendation of AAMI, these heartbeats are further divided into five heartbeat classes. Table 2 shows the mapping of AAMI classes and MIT-BIH arrhythmia heartbeat types. Since Q is practically non-existent, we ignore it like others (*Mar et al., 2011*; *Zhang et al., 2014*). Meanwhile, four recordings with paced beats are

**Table 3 Detailed breakdown of the dataset.**

| Dataset | No. of samples per AAMI class | | | | Total |
|---|---|---|---|---|---|
| | N | SVEB | VEB | F | |
| DS1 | 45,824 | 943 | 3,788 | 414 | 50,969 |
| DS2 | 44,218 | 1,836 | 3,219 | 388 | 49,661 |
| Total (DS1 + DS2) | 90,042 | 2,779 | 7,007 | 802 | 100,630 |

removed in consistent with the AAMI recommended practice, namely 102, 104, 107 and 217. Since all records have ML II ECG signals and they are widely used in wireless body sensor network (WBSN) based ECG applications, this lead ECG signal is used for heartbeat classification in the article.

As we mentioned in related works, the article focuses on the heartbeat classification under the inter-patient paradigm. To facilitate comparison with existing works, we follow *De Chazal, O'Dwyer & Reilly (2004)* to split the dataset into two subsets. Each contains regular and complex arrhythmia records and has roughly the same number of heartbeat types. Table 3 shows the details of two subsets. The first (DS1) is used for training whereas the second (DS2) is used to test the heartbeat classification performance (*De Chazal, O'Dwyer & Reilly, 2004*). No patient appears in both subsets at the same time.

## Model training and performance metrics

In the study, the general focal loss (non-α-balanced focal loss) is used as the loss function, and the modulating factor γ is set to the default value (γ = 2). Since Adam can accelerate the model training, we use it as the optimizer. The batch size of the model is set to 512 and the maximum epoch is 50. The initial learning rate is 0.001, and reduced by 0.1 times every 10 epochs. In addition, in order to avoid overfitting, the l2 penalty is set to 1e−3 based on trial and error. The model is implemented using Keras and trained on the NVIDIA GeForce RTX 2080Ti graphical processing unit.

To evaluate the performance of our proposed method, three widely used metrics are adopted, namely positive predictive value (PPV), sensitivity (SE), and accuracy (ACC), which are defined as:

$$PPV_i = \frac{TP_i}{TP_i + FP_i}$$

$$SE_i = \frac{TP_i}{TP_i + FN_i}$$

$$ACC_i = \frac{TP_i + TN_i}{TP_i + TN_i + FP_i + FN_i}$$

where $TP_i$ (true positive) refers to the number of the *ith* class is correctly classified, $FP_i$ (false positive) is equal to the number of heartbeats misclassified as the *ith* class, $TN_i$ (true negative) is the number of heartbeats that are not in the *ith* class and not classified into the *ith* class, and $FN_i$ (false negative) is equal to the number of heartbeats of the *ith* class classified as other classes. $PPV_i$ indicates the proportion of positive correct

**Table 4 Performance comparison of our proposed method with existing works in SVEB and VEB classes.**

| Methods | SVEB | | | | VEB | | | |
|---|---|---|---|---|---|---|---|---|
| | PPV (%) | SE (%) | F1 (%) | Accuracy (%) | PPV (%) | SE (%) | F1 (%) | Accuracy (%) |
| De Chazal, O'Dwyer & Reilly (2004) | 38.53 | 75.98 | 51.13 | 94.61 | 81.67 | 80.31 | 80.98 | 97.62 |
| Chen et al. (2017) | 38.40 | 29.50 | 33.36 | 95.34 | 85.25 | 70.85 | 77.38 | 97.32 |
| Zhang et al. (2014) | 35.98 | 79.06 | 49.46 | 93.33 | 92.75 | 85.48 | 88.96 | 98.63 |
| Mar et al. (2011) | 33.53 | 83.22 | 47.80 | 93.28 | 75.89 | 86.75 | 80.96 | 97.35 |
| Liu et al. (2019) | 39.87 | 33.12 | 36.18 | 95.49 | 76.51 | 90.20 | 82.79 | 97.45 |
| Garcia et al. (2017) | 53.00 | 62.00 | 57.15 | – | 59.40 | 87.30 | 70.70 | – |
| Our proposed method | 68.34 | 81.37 | 74.29 | 97.92 | 91.12 | 93.72 | 92.40 | 99.00 |

classification, and $SE_i$ reflects the sensitivity of the classifier in the $ith$ class. $ACC_i$ is the ratio of all correct classifications.

Since the heartbeat classes are imbalanced, f1-score (F1) is also selected as the performance measure, defined as:

$$F1_i = \frac{2 \times PPV_i \times SE_i}{PPV_i + SE_i}$$

f1-score takes both the positive predictive value $PPV_i$ and sensitivity $SE_i$ into account, and is generally useful than $ACC_i$ in the imbalance class distribution (*Chen, 2009*).

## Comparison with existing works

Based on *De Chazal, O'Dwyer & Reilly (2004)*, the dataset is divided into DS1 and DS2 datasets. DS1 is used for training and DS2 is used to test our proposed method. For fair evaluation, we compare works (*Chen et al., 2017*; *De Chazal, O'Dwyer & Reilly, 2004*; *Garcia et al., 2017*; *Liu et al., 2019*; *Mar et al., 2011*; *Zhang et al., 2014*) that adopt the same strategy. As SVEB and VEB are more important than other classes, we list the detailed information of these two classes in Table 4. The experimental results show that the proposed method has better recognition in the inter-patient paradigm, with F1s of SVEB and VEB of 74.29% and 92.40%, respectively. In particular, the PPV of SVEB is 68.34%, indicating that the proposed method has better SVEB recognition ability. The 93.72% SE of VEB is superior to most reported works. The evaluation results of all four-classes are listed in Table 5. The results related to PPV and SE are close to or surpass those obtained with existing works except for *F*. For category *F*, it is mainly composed of the fusion of ventricular beat and normal beat, which is very close to the normal heartbeat. Meanwhile, compared with other categories, *F* has the least number and the most serious imbalance. As a result, the performance of existing works is unstable in this category, usually a large number of *N* is predicted as *F* or *F* is predicted as *N*. In the article, although the focus loss is introduced, due to the high imbalance of *F*, the proposed method cannot extract the discriminate features. *Mar et al. (2011)* although, obtains the best PPV in *F*, a large number of *N* is incorrectly classified as *F*. We suggest that category *F* can be included in other categories in future research.

**Table 5 Performance comparison of our proposed method with existing works in all four classes.**

| Methods | Accuracy[a] | Macro-F1[b] | N | | SVEB | | VEB | | F | |
|---|---|---|---|---|---|---|---|---|---|---|
| | | | PPV (%) | SE (%) | PPV (%) | SE (%) | PPV (%) | SE (%) | PPV (%) | SE (%) |
| *De Chazal, O'Dwyer & Reilly (2004)* | 86.24 | 60.12 | 99.17 | 87.06 | 38.53 | 75.98 | 81.67 | 80.31 | 8.57 | 89.43 |
| *Chen et al. (2017)* | 93.14 | 51.91 | 95.42 | 98.42 | 38.40 | 29.50 | 85.25 | 70.85 | 0.00 | 0.00 |
| *Zhang et al. (2014)* | 88.34 | 64.02 | 98.98 | 88.94 | 35.98 | 79.06 | 92.75 | 85.48 | 13.73 | 93.81 |
| *Mar et al. (2011)* | 88.99 | 62.24 | 99.12 | 89.64 | 33.53 | 83.22 | 75.89 | 86.75 | 16.57 | 61.08 |
| *Liu et al. (2019)* | – | 58.50 | 96.66 | 94.06 | 39.87 | 33.12 | 76.51 | 90.20 | 12.99 | 40.72 |
| *Garcia et al. (2017)* | 92.40 | 55.95 | 98.00 | 94.00 | 53.00 | 62.00 | 59.40 | 87.30 | – | – |
| Our proposed method | 92.53 | 66.09 | 98.20 | 93.67 | 68.34 | 81.37 | 91.12 | 93.72 | 1.06 | 5.41 |

**Notes:**
[a] Accuracy = $(TP_N + TP_{SVEB} + TP_{VEB} + TP_F)$/number of testing heartbeats.
[b] Average f1-score of four AAMI classes.

**Table 6 Performance comparison of focal loss and cross-entropy loss.**

| Methods | AAMI class | Performance metrics | | | |
|---|---|---|---|---|---|
| | | PPV (%) | SE (%) | F1 (%) | Accuracy (%) |
| Focal Loss | *N* | 98.20 | 93.67 | 95.88 | 92.84 |
| | SVEB | 68.34 | 81.37 | 74.29 | 97.92 |
| | VEB | 91.12 | 93.72 | 92.40 | 99.00 |
| | *F* | 1.06 | 5.41 | 1.77 | 95.31 |
| | Average[a] | 64.68 | 68.55 | 66.09 | 96.27 |
| Cross-Entropy Loss | *N* | 98.11 | 92.42 | 95.18 | 91.67 |
| | SVEB | 57.11 | 80.72 | 66.89 | 97.05 |
| | VEB | 83.95 | 93.45 | 88.44 | 98.42 |
| | *F* | 0.11 | 0.52 | 0.18 | 95.54 |
| | Average[a] | 59.82 | 66.77 | 62.67 | 95.67 |

**Note:**
[a] Refers to the average value of the corresponding metrics of four AAMI classes.

## DISCUSSION

### Focal loss vs. Cross-entropy loss

Since the heartbeat has an imbalanced class distribution, the cross-entropy loss is replaced by the focal loss as the loss function of the model in the article. The performance comparison of the two losses is listed in Table 6. Both losses have similar overall accuracy, but compared to cross-entropy loss, the overall PPV, SE and F1 of the focal loss are significantly improved. An overall PPV of 64.68%, SE of 68.55%, and F1 of 66.09% are achieved by the focal loss, while the cross-entropy loss obtains an overall PPV of 59.82%, SE of 66.77%, and F1 of 62.67%. The corresponding metrics increased by 4.86%, 1.78%, and 3.42%, respectively. In addition, for each specific class, the PPV, SE, and F1 of the focal loss also have achieved comparable or better performance than the cross-entropy loss, especially in the F1.

The confusion matrix of the two losses is listed in Table 7. The focal loss achieves a total of 45,952 correct predictions, while the cross-entropy obtains 45,358 correct

Table 7 Confusion matrix of focal loss and cross-entropy loss.

| | Focal loss | Predicted class | | | | Total | Cross-entropy loss | Predicted class | | | | Total |
|---|---|---|---|---|---|---|---|---|---|---|---|---|
| | | N | SVEB | VEB | F | | | N | SVEB | VEB | F | |
| Ground Truth | N | 41,420 | 671 | 206 | 1,921 | 44,218 | N | 40,866 | 1,082 | 462 | 1,808 | 44,218 |
| | SVEB | 282 | 1,494 | 57 | 3 | 1,836 | SVEB | 246 | 1,482 | 95 | 13 | 1,836 |
| | VEB | 141 | 21 | 3,017 | 40 | 3,219 | VEB | 173 | 31 | 3,008 | 7 | 3,219 |
| | F | 336 | 0 | 31 | 21 | 388 | F | 368 | 0 | 18 | 2 | 388 |
| | Total | 42,179 | 2,186 | 3,311 | 1,985 | 49,661 | Total | 41,653 | 2,595 | 3583 | 1,830 | 49,661 |

predictions. With the focal loss, the total correct prediction has slightly increased, perhaps due to the suppression of easy-to-classify samples by focal loss.

## CONCLUSIONS

A hybrid method for heartbeat classification via CNN, MLP and focal loss is developed in the article. Among them, CNN is used to extract the morphological features of the heartbeat. Then the morphological features are combined with the RR intervals features and input into the MLP to perform heartbeat classification. Furthermore, in order to avoid the impact of heartbeat imbalance, a focal loss function is introduced. Tested by using the MIT-BIH arrhythmia database, the experimental results confirm that the method has good overall performance, with F1 of 66.09% and accuracy of 96.27%. The superiority of the proposed method is due to multifactorial: (I) Compared with traditional hand-craft features, CNN as an automatic extraction method can adapt to small mutations in ECG signals to obtain powerful features; (II) Besides the CNN-based morphological features, the pre-extracted RR interval features are also combined to build the model, avoiding the loss of dynamic information due to heartbeat segmentation; (III) A focal loss function is introduced to solve the class imbalance, preventing the model from biasing towards the majority class; (IV) One-step training can improve the model to obtain better feature abstraction capabilities. Due to the simple yet effective of the proposed inter-patient method, it has the potential to be used for personal automatic heartbeat classification for surveillance in telemedicine.

The encouraging results have inspired continuous exploration. The future work will include (I) testing the performance of the developed model with more ECG signals; (II) designing or modifying CNN architecture to further improve the performance of our method; (III) trying to use additional techniques such as wavelet transform to convert time-domain information to frequency-domain information to reduce the difficulty of CNN feature extraction.

### Funding

The work is supported by the Science and Technology Service Network Initiative of the Chinese Academy of Sciences (Grant No. KFJ-STS-ZDTP-079). The funders had no role in

study design, data collection and analysis, decision to publish, or preparation of the manuscript.

## Grant Disclosures
The following grant information was disclosed by the authors:
Science and Technology Service Network Initiative of the Chinese Academy of Sciences: KFJ-STS-ZDTP-079.

## Competing Interests
Mei Yang is employed by Beijing Huaru Technology Co., Ltd.

## Author Contributions
- Tao Wang conceived and designed the experiments, performed the experiments, analyzed the data, performed the computation work, prepared figures and/or tables, authored or reviewed drafts of the paper, and approved the final draft.
- Changhua Lu conceived and designed the experiments, authored or reviewed drafts of the paper, and approved the final draft.
- Mei Yang conceived and designed the experiments, performed the experiments, authored or reviewed drafts of the paper, and approved the final draft.
- Feng Hong conceived and designed the experiments, analyzed the data, performed the computation work, authored or reviewed drafts of the paper, and approved the final draft.
- Chun Liu conceived and designed the experiments, authored or reviewed drafts of the paper, and approved the final draft.

## Data Availability
This work uses a public dataset from:

Goldberger, A., Amaral, L., Glass, L., Hausdorff, J., Ivanov, P.C., Mark, R., Mietus, J.E., Moody, G.B., Peng, C.K. and Stanley, H.E., 2000. PhysioBank, PhysioToolkit, and PhysioNet: Components of a new research resource for complex physiologic signals. Circulation [Online]. 101 (23), pp. e215–e220. https://www.physionet.org/content/mitdb/1.0.0/.

Code is available at GitHub: https://github.com/JackAndCole/Deep-Neural-Network-For-Heartbeat-Classification.

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
