# Peer review of "A hybrid method for heartbeat classification via convolutional neural networks, multilayer perceptrons and focal loss"

_PeerJ Computer Science, doi:10.7717/peerj-cs.324_

## Round 0.1 · original submission · Major Revisions

The authors should improve the presentation of the paper and properly describe the novelty, contribution and motivation. The proposed solution must be compared with other similar solutions. Conclusions should highlight the advantages/disadvantages of the present method with respect to similar works.

Reviewer 1 ·

Basic reporting

Clear English is used throughout. Literature references are sufficient. Professional article structure, figures and tables are provided.

Experimental design

1. Since from fig. 1 the first part of your model is cnn, therefore the question about heart beat signal is important. What is the input to your cnn model?
2. The training of the system was not defined. Discuss how the model was trained and show advances of your approach.

Validity of the findings

1. Please provide samples of morphological features on cnn output, and also show how the signal looks during cnn processing.

Additional comments

1. Why the model is called deep neural network? Actually there is not much relation to models sourced in deep learning, and giving such name just because of use of cnn+nn is not good. I suggest to revise the title to sth what shows a hybrid approach of
2. Please provide samples of morphological features on cnn output, and also show how the signal looks during cnn processing. these two structures.

Reviewer 2 ·

Basic reporting

The document is well structured, the raw data is provided and the results are clear. However, there are several issues that need to be addressed before publication.

One of the most important issue is that authors need to provide greater justification for their study and better background research. In particular, the related work section should include more similar research. There are several previous research studies that use CNN to classify arrhythmia and others that use the focal loss in heartbeat classification to address the problem of the imbalance classes. Authors should cite these papers and highlight the advantages/differences of their own work with respect to previous work. For example, the paper "Detection and classification of arrhythmias at the cardiologist level in ambulatory electrocardiograms using a deep neural network" which is the state of the art in the field, or other papers such as "Detection of arrhythmias using a deep convolutional neural network with long-lasting ECG signals", "The classification of the electrocardiogram beats based on a deep convolutional neural network and the focal loss” (https://doi.org/10.1016/j.compbiomed.2020.103866). In addition, the comparison of the results should be done with more recent work. For instance, the reference "Mondéjar-Guerra V, Novo J, Rouco J, Penedo MG, Ortega MJBSP, and Control. 2019" uses the same dataset and train/test procedure than the present paper.

The statement "Since CNN has better noise immunity" should come back with a citation.

The next important issue is the paper formulas. They must have a consistent notation to improve readers' understanding. The loss of cross entropy has a minus sign before the sum. The same is true for the formulas for FL and FL'. Also the pt and pi in the formulas for FL and FL' are the same thing. When the authors explain the convolution, the letters for the size of the filter, the output of the convolutional layer, etc., must be the same in all formulas. Please rewrite all formulas correctly.

In the sentence "Due to variations between patients in the ECG signal, the RR interval of different patients cannot be directly compared, in this work we subtract the average from the patient's RR interval to eliminate the effect". It is not clear what you mean by the average of the patient's RR interval. Do you use the entire patient's time series to calculate the average? Do you subtract the average of all the RR characteristics? Please be more specific.

Finally, the authors state "We adopt a multilayer perceptron (MLP) classifier because it can form an end-to-end classification with CNN, that is, feature extraction and classification can be performed simultaneously." However, RR features are preextracted, so not all feature extraction can be performed simultaneously with their method. Please clarify this point in the text.

Some minor comments:
1. The English language should be improved. Mainly, there are some grammar mistakes , for instance:
a. The output of the convolution layer IS calculateD as follows:
b. is one of the MOST widely used activation functions
c. BE a mini-bacth of the entire training set, the output of the batch normalization IS as follows
d. The output layer neuron is equal to the predicted classes, and in the paper the neuron is 4. Please rephrase the part” and in the paper the neuron is 4”
e. The cross-entropy loss is THE most widely used in deep neural network classification
f. BE the ground truth vector

2. Please also consider:
a. Rephrase this sentence with a full point “ The proposed method adopts both the morphological characteristics and dynamic information of the heartbeat at the same time. THE morphological characteristics are extracted through a convolutional neural network.”
b. Althoug it is mentioned later in the text, I recommend to clarify in the introduction that the authors use a 1 dimensional CNN.
c.The time interval between two consecutive R-peaks is normally called the RR interval (Ruangsuwana et al. 2010), which represents the dynamic information of the heartbeat. consider replacing "represents " with a word that fits better

Experimental design

Due to the lack of a good related work, it is not clear how the present research fills an identified knoledge gap.

The methods are described with sufficient detail and information can be replicate.

Validity of the findings

Conclusions should highligth the advantages/disadvantages of the present method respect to similar works.

---

## Round 0.2 · Minor Revisions

The authors should faithfully address the comments of the reviewer when revising their manuscript.

Reviewer 1 ·

Basic reporting

the paper is revised

Experimental design

the paper is revised

Validity of the findings

the paper is revised

Additional comments

the paper is revised

Reviewer 2 ·

Basic reporting

Minor comments:

1) "Although the performance of heartbeat classification is improved, these works mainly focus on using CNN to extract the heartbeat morphological features. While arrhythmia not only changes the morphology of the heartbeat but also causes abnormal RR intervals". Please re-elaborate. The main focus of the classification is to improve the classification no matter how. You can argue that CNN would be improved/simplified by integrating the RR

2)"Two median filters are combined to REMOVE THE BASELINE WANDERING of the ECG signal in the paper. First, the QRS complexes and P-waves are removed using a 200-ms width median filter, and then a 600-ms width".

3) "the hearbeat CLASSES ARE imbalanced" not the hearbeats.

4) Although the authors have improved the formulas, the CNN part is still a bit confusing. The convolution formula (line 187) should say that m is the size of the filter. This is mentioned but later in the text.

5) "Let t_i and p_i BE the ground truth and the estimated probability for each category i in C, the cross-entropy loss is computed by". What is C?

6) "In particular, the PPV of SVEB is 68.34%, indicating that the proposed method has a strong capability to recognize SVEB" No. It is better than the other methods listed in the table, but a PPV of about 70% cannot be qualified as a strong capability.



7) Grammar is better, but there is still room for improvement. The authors should carefully review the manuscript. For example:

- "Let t_i and p_i BE the ground truth and the estimated probability for each category i in C, the cross-entropy loss is computed by".
- Then the morphological features are combined with the RR intervals features AND input into the MLP to perform heartbeat classification.
- The kernel of the convolution is reduced as the network structure becomes deeper.
- I do not understand this sentence "The results related to PPV and SE are close to or surpass those obtained with existing works expect for F. Accuracy and Marco-F1 reached 92.53% and 66.09%, respectively. "
- For clarity consider including a full point in this sentence. "Fig. 1 shows the overall structure of the proposed method. The proposed method includes three steps: ECG denoising, feature extraction, and classification. THE the feature extraction step contains RR intervals features extraction and morphological features extraction via CNN architecture".

Experimental design

The methods are described with sufficient detail and information to replicated.

Validity of the findings

Authors should provide some background in the text on the results obtained with their method for class F, which are much lower than those obtained with the other methods. Any idea why it works worse? The authors claim that they used focal loss rather than loss of cross entropy to better address unbalanced class distribution.

---

## Round 0.3 · accepted · Accept

Following the latest round of revisions, I have decided to accept this article for publication.